# Loss-aware Weight Quantization of Deep Networks

**Lu Hou, James T. Kwok**
Department of Computer Science and Engineering
Hong Kong University of Science and Technology
Hong Kong
{lhouab, jamesk}@cse.ust.hk

## Abstract

The huge size of deep networks hinders their use in small computing devices. In this paper, we consider compressing the network by weight quantization. We extend a recently proposed loss-aware weight binarization scheme to ternarization, with possibly different scaling parameters for the positive and negative weights, and $m$-bit (where $m > 2$) quantization. Experiments on feedforward and recurrent neural networks show that the proposed scheme outperforms state-of-the-art weight quantization algorithms, and is as accurate (or even more accurate) than the full-precision network.

## 1 Introduction

The last decade has witnessed huge success of deep neural networks in various domains. Examples include computer vision, speech recognition, and natural language processing (LeCun et al., 2015). However, their huge size often hinders deployment to small computing devices such as cell phones and the internet of things. Many attempts have been recently made to reduce the model size. One common approach is to prune a trained dense network (Han et al., 2015; 2016). However, most of the pruned weights may come from the fully-connected layers where computations are cheap, and the resultant time reduction is insignificant. Li et al. (2017b) and Molchanov et al. (2017) proposed to prune filters in the convolutional neural networks based on their magnitudes or significance to the loss. However, the pruned network has to be retrained, which is again expensive.

Another direction is to use more compact models. GoogleNet (Szegedy et al., 2015) and ResNet (He et al., 2016) replace the fully-connected layers with simpler global average pooling. However, they are also deeper. SqueezeNet (Iandola et al., 2016) reduces the model size by replacing most of the $3 \times 3$ filters with $1 \times 1$ filters. This is less efficient on smaller networks because the dense $1 \times 1$ convolutions are costly. MobileNet (Howard et al., 2017) compresses the model using separable depth-wise convolution. ShuffleNet (Zhang et al., 2017) utilizes pointwise group convolution and channel shuffle to reduce the computation cost while maintaining accuracy. However, highly optimized group convolution and depth-wise convolution implementations are required. Alternatively, Novikov et al. (2015) compressed the model by using a compact multilinear format to represent the dense weight matrix. The CP and Tucker decompositions have also been used on the kernel tensor in CNNs (Lebedev et al., 2014; Kim et al., 2016). However, they often need expensive fine-tuning.

Another effective approach to compress the network and accelerate training is by quantizing each full-precision weight to a small number of bits. This can be further divided to two sub-categories, depending on whether pre-trained models are used (Lin et al., 2016a; Mellempudi et al., 2017) or the quantized model is trained from scratch (Courbariaux et al., 2015; Li et al., 2017a). Some of these also directly learn with low-precision weights, but they usually suffer from severe accuracy deterioration (Li et al., 2017a; Miyashita et al., 2016). By keeping the full-precision weights during learning, Courbariaux et al. (2015) pioneered the BinaryConnect algorithm, which uses only one bit for each weight while still achieving state-of-the-art classification results. Rastegari et al. (2016) further incorporated weight scaling, and obtained better results. Instead of simply finding the closest binary approximation of the full-precision weights, a loss-aware scheme is proposed in (Hou et al., 2017). Beyond binarization, TernaryConnect (Lin et al., 2016b) quantizes each weight to

$\{-1, 0, 1\}$. Li & Liu (2016) and Zhu et al. (2017) added scaling to the ternarized weights, and DoReFa-Net (Zhou et al., 2016) further extended quantization to more than three levels. However, these methods do not consider the effect of quantization on the loss, and rely on heuristics in their procedures (Zhou et al., 2016; Zhu et al., 2017). Recently, a loss-aware low-bit quantized neural network is proposed in (Leng et al., 2017). However, it uses full-precision weights in the forward pass and the extra-gradient method (Vasilyev et al., 2010) for update, both of which are expensive.

In this paper, we propose an efficient and disciplined ternarization scheme for network compression. Inspired by (Hou et al., 2017), we explicitly consider the effect of ternarization on the loss. This is formulated as an optimization problem which is then solved efficiently by the proximal Newton algorithm. When the loss surface's curvature is ignored, the proposed method reduces to that of (Li & Liu, 2016), and is also related to the projection step of (Leng et al., 2017). Next, we extend it to (i) allow the use of different scaling parameters for the positive and negative weights; and (ii) the use of $m$ bits (where $m > 2$) for weight quantization. Experiments on both feedforward and recurrent neural networks show that the proposed quantization scheme outperforms state-of-the-art algorithms.

**Notations:** For a vector $\mathbf{x}$, $\sqrt{\mathbf{x}}$ denotes the element-wise square root (i.e., $[\sqrt{\mathbf{x}}]_i = \sqrt{x_i}$), $|\mathbf{x}|$ is the element-wise absolute value, $\|\mathbf{x}\|_p = (\sum_i |x_i|^p)^{\frac{1}{p}}$ is its $p$-norm, and $\text{Diag}(\mathbf{x})$ returns a diagonal matrix with $\mathbf{x}$ on the diagonal. For two vectors $\mathbf{x}$ and $\mathbf{y}$, $\mathbf{x} \odot \mathbf{y}$ denotes the element-wise multiplication and $\mathbf{x} \oslash \mathbf{y}$ the element-wise division. Given a threshold $\Delta$, $\mathbf{I}_\Delta(\mathbf{x})$ returns a vector such that $[\mathbf{I}_\Delta(\mathbf{x})]_i = 1$ if $x_i > \Delta$, $-1$ if $x_i < -\Delta$, and 0 otherwise. $\mathbf{I}_\Delta^+(\mathbf{x})$ considers only the positive threshold, i.e., $[\mathbf{I}_\Delta^+(\mathbf{x})]_i = 1$ if $x_i > \Delta$, and 0 otherwise. Similarly, $[\mathbf{I}_\Delta^-(\mathbf{x})]_i = -1$ if $x_i < -\Delta$, and 0 otherwise. For a matrix $\mathbf{X}$, $\text{vec}(\mathbf{X})$ returns a vector by stacking all the columns of $\mathbf{X}$, and $\text{diag}(\mathbf{X})$ returns a vector whose entries are from the diagonal of $\mathbf{X}$.

## 2 RELATED WORK

Let the full-precision weights from all $L$ layers be $\mathbf{w} = [\mathbf{w}_1^\top, \mathbf{w}_2^\top, \ldots, \mathbf{w}_L^\top]^\top$, where $\mathbf{w}_l = \text{vec}(\mathbf{W}_l)$, and $\mathbf{W}_l$ is the weight matrix at layer $l$. The corresponding quantized weights will be denoted $\hat{\mathbf{w}} = [\hat{\mathbf{w}}_1^\top, \hat{\mathbf{w}}_2^\top, \ldots, \hat{\mathbf{w}}_L^\top]^\top$.

### 2.1 WEIGHT BINARIZED NETWORKS

In BinaryConnect (Courbariaux et al., 2015), each element of $\mathbf{w}_l$ is binarized to $-1$ or $+1$ by using the sign function: $\text{Binarize}(\mathbf{w}_l) = \text{sign}(\mathbf{w}_l)$. In the Binary-Weight-Network (BWN) (Rastegari et al., 2016), a scaling parameter is also included, i.e., $\text{Binarize}(\mathbf{w}_l) = \alpha_l \mathbf{b}_l$, where $\alpha_l > 0$, $\mathbf{b}_l \in \{-1, +1\}^{n_l}$ and $n_l$ is the number of weights in $\mathbf{w}_l$. By minimizing the difference between $\mathbf{w}_l$ and $\alpha_l \mathbf{b}_l$, the optimal $\alpha_l, \mathbf{b}_l$ have the simple form: $\alpha_l = \|\mathbf{w}_l\|_1/n_l$, and $\mathbf{b}_l = \text{sign}(\mathbf{w}_l)$.

Instead of simply finding the best binary approximation for the full-precision weight $\mathbf{w}_l^t$ at iteration $t$, the loss-aware binarized network (LAB) directly minimizes the loss w.r.t. the binarized weight $\alpha_l^t \mathbf{b}_l^t$ (Hou et al., 2017). Let $\mathbf{d}_l^{t-1}$ be a vector containing the diagonal of an approximate Hessian of the loss. It can be shown that $\alpha_l^t = \|\mathbf{d}_l^{t-1} \odot \mathbf{w}_l^t\|_1/\|\mathbf{d}_l^{t-1}\|_1$ and $\mathbf{b}_l^t = \text{sign}(\mathbf{w}_l^t)$.

### 2.2 WEIGHT TERNARIZED NETWORKS

In a weight ternarized network, zero is used as an additional quantized value. In TernaryConnect (Lin et al., 2016b), each weight value is clipped to $[-1, 1]$ before quantization, and then a non-negative weight $[\mathbf{w}_l^t]_i$ is stochastically quantized to 1 with probability $[\mathbf{w}_l^t]_i$ (and 0 otherwise). When $[\mathbf{w}_l^t]_i$ is negative, it is quantized to $-1$ with probability $-[\mathbf{w}_l^t]_i$, and 0 otherwise.

In the ternary weight network (TWN) (Li & Liu, 2016), $\mathbf{w}_l^t$ is quantized to $\hat{\mathbf{w}}_l^t = \alpha_l^t \mathbf{I}_{\Delta_l^t}(\mathbf{w}_l^t)$, where $\Delta_l^t$ is a threshold (i.e., $[\hat{\mathbf{w}}_l^t]_i = \alpha_l^t$ if $[\mathbf{w}_l^t]_i > \Delta_l^t$, $-\alpha_l^t$ if $[\mathbf{w}_l^t]_i < -\Delta_l^t$ and 0 otherwise). To obtain $\Delta_l^t$ and $\alpha_l^t$, TWN minimizes the $\ell_2$-distance between the full-precision and ternarized

weights, leading to

$$\Delta_l^t = \arg\max_{\Delta>0} \frac{1}{\|\mathbf{I}_\Delta(\mathbf{w}_l^t)\|_1} \left( \sum_{i:|[\mathbf{w}_l^t]_i|>\Delta_l^t} |[\mathbf{w}_l^t]_i| \right)^2, \quad \alpha_l^t = \frac{1}{\|\mathbf{I}_{\Delta_l^t}(\mathbf{w}_l^t)\|_1} \sum_{i:|[\mathbf{w}_l^t]_i|>\Delta_l^t} |[\mathbf{w}_l^t]_i|. \quad (1)$$

However, $\Delta_l^t$ in (1) is difficult to solve. Instead, TWN simply sets $\Delta_l^t = 0.7 \cdot \mathbf{E}(|\mathbf{w}_l^t|)$ in practice.

In TWN, one scaling parameter ($\alpha_l^t$) is used for both the positive and negative weights at layer $l$. In the trained ternary quantization (TTQ) network (Zhu et al., 2017), different scaling parameters ($\alpha_l^t$ and $\beta_l^t$) are used. The weight $\mathbf{w}_l^t$ is thus quantized to $\hat{\mathbf{w}}_l^t = \alpha_l^t \mathbf{I}_{\Delta_l^t}^+(\mathbf{w}_l^t) + \beta_l^t \mathbf{I}_{\Delta_l^t}^-(\mathbf{w}_l^t)$. The scaling parameters are learned by gradient descent. As for $\Delta_l^t$, two heuristics are used. The first sets $\Delta_l^t$ to a constant fraction of $\max(|\mathbf{w}_l^t|)$, while the second sets $\Delta_l^t$ such that at all layers are equally sparse.

## 2.3 WEIGHT QUANTIZED NETWORKS

In a weight quantized network, $m$ bits (where $m \geq 2$) are used to represent each weight. Let $\mathcal{Q}$ be a set of $(2k+1)$ quantized values, where $k = 2^{m-1} - 1$. The two popular choices of $\mathcal{Q}$ are $\left\{-1, -\frac{k-1}{k}, \ldots, -\frac{1}{k}, 0, \frac{1}{k}, \ldots, \frac{k-1}{k}, 1\right\}$ (linear quantization), and $\left\{-1, -\frac{1}{2}, \ldots, -\frac{1}{2^{k-1}}, 0, \frac{1}{2^{k-1}}, \ldots, \frac{1}{2}, 1\right\}$ (logarithmic quantization). By limiting the quantized values to powers of two, logarithmic quantization is advantageous in that expensive floating-point operations can be replaced by cheaper bit-shift operations. When $m = 2$, both schemes reduce to $\mathcal{Q} = \{-1, 0, 1\}$.

In the DoReFa-Net (Zhou et al., 2016), weight $\mathbf{w}_l^t$ is heuristically quantized to $m$-bit, with:[1]

$$[\hat{\mathbf{w}}_l^t]_i = 2 \cdot \text{quantize}_m \left( \frac{\tanh([\mathbf{w}_l^t]_i)}{2\max(|\tanh([\mathbf{w}_l^t]_i)|)} + \frac{1}{2} \right) - 1$$

in $\left\{-1, -\frac{2^m-2}{2^m-1}, \ldots, -\frac{1}{2^m-1}, \frac{1}{2^m-1}, \ldots, \frac{2^m-2}{2^m-1}, 1\right\}$, where $\text{quantize}_m(x) = \frac{1}{2^m-1}\text{round}((2^m - 1)x)$. Similar to loss-aware binarization (Hou et al., 2017), Leng et al. (2017) proposed a loss-aware quantized network called low-bit neural network (LBNN). The alternating direction method of multipliers (ADMM) (Boyd et al., 2011) is used for optimization. At the $t$th iteration, the full-precision weight $\mathbf{w}_l^t$ is first updated by the method of extra-gradient (Vasilyev et al., 2010):

$$\tilde{\mathbf{w}}_l^t = \mathbf{w}_l^{t-1} - \eta^t \nabla_l \mathcal{L}(\mathbf{w}_l^{t-1}), \quad \mathbf{w}_l^t = \mathbf{w}_l^{t-1} - \eta^t \nabla_l \mathcal{L}(\tilde{\mathbf{w}}_l^t), \quad (2)$$

where $\mathcal{L}$ is the augmented Lagrangian in the ADMM formulation, and $\eta^t$ is the stepsize. Next, $\mathbf{w}_l^t$ is projected to the space of $m$-bit quantized weights so that $\hat{\mathbf{w}}_l^t$ is of the form $\alpha_l \mathbf{b}_l$, where $\alpha_l > 0$, and $\mathbf{b}_l \in \left\{-1, -\frac{1}{2}, \ldots, -\frac{1}{2^{k-1}}, 0, \frac{1}{2^{k-1}}, \ldots, \frac{1}{2}, 1\right\}$.

## 3 LOSS-AWARE QUANTIZATION

### 3.1 TERNARIZATION USING PROXIMAL NEWTON ALGORITHM

In weight ternarization, TWN simply finds the closest ternary approximation of the full precision weight at each iteration, while TTQ sets the ternarization threshold heuristically. Inspired by LAB (for binarization), we consider the loss explicitly during quantization and obtain the quantization thresholds and scaling parameter by solving an optimization problem.

As in TWN, the weight $\mathbf{w}_l$ is ternarized as $\hat{\mathbf{w}}_l = \alpha_l \mathbf{b}_l$, where $\alpha_l > 0$ and $\mathbf{b}_l \in \{-1, 0, 1\}^{n_l}$. Given a loss function $\ell$, we formulate weight ternarization as the following optimization problem:

$$\min_{\hat{\mathbf{w}}} \ell(\hat{\mathbf{w}}) : \hat{\mathbf{w}}_l = \alpha_l \mathbf{b}_l, \ \alpha_l > 0, \ \mathbf{b}_l \in \mathcal{Q}^{n_l}, \ l = 1, \ldots, L, \quad (3)$$

where $\mathcal{Q}$ is the set of desired quantized values. As in LAB, we will solve this using the proximal Newton method (Lee et al., 2014; Rakotomamonjy et al., 2016). At iteration $t$, the objective is replaced by the second-order expansion

$$\ell(\hat{\mathbf{w}}^{t-1}) + \nabla\ell(\hat{\mathbf{w}}^{t-1})^\top(\hat{\mathbf{w}} - \hat{\mathbf{w}}^{t-1}) + \frac{1}{2}(\hat{\mathbf{w}} - \hat{\mathbf{w}}^{t-1})^\top \mathbf{H}^{t-1}(\hat{\mathbf{w}} - \hat{\mathbf{w}}^{t-1}), \quad (4)$$

---

[1]Note that the quantized value of 0 is not used in DoReFa-Net.

where $\mathbf{H}^{t-1}$ is an estimate of the Hessian of $\ell$ at $\hat{\mathbf{w}}^{t-1}$. We use the diagonal equilibration preconditioner (Dauphin et al., 2015), which is robust in the presence of saddle points and also readily available in popular stochastic deep network optimizers such as Adam (Kingma & Ba, 2015). Let $\mathbf{D}_l$ be the approximate diagonal Hessian at layer $l$. We use $\mathbf{D} = \mathrm{Diag}([\mathrm{diag}(\mathbf{D}_1)^{\top}, \ldots, \mathrm{diag}(\mathbf{D}_L)^{\top}]^{\top})$ as an estimate of $\mathbf{H}$. Substituting (4) into (3), we solve the following subproblem at the $t$th iteration:

$$\min_{\hat{\mathbf{w}}^t} \quad \nabla\ell(\hat{\mathbf{w}}^{t-1})^{\top}(\hat{\mathbf{w}}^t - \hat{\mathbf{w}}^{t-1}) + \frac{1}{2}(\hat{\mathbf{w}}^t - \hat{\mathbf{w}}^{t-1})^{\top}\mathbf{D}^{t-1}(\hat{\mathbf{w}}^t - \hat{\mathbf{w}}^{t-1}) \tag{5}$$

$$\text{s.t.} \quad \hat{\mathbf{w}}_l^t = \alpha_l^t \mathbf{b}_l^t, \ \alpha_l^t > 0, \ \mathbf{b}_l^t \in \mathcal{Q}^{n_l}, \ l = 1, \ldots, L.$$

**Proposition 3.1** *The objective in (5) can be rewritten as*

$$\min_{\hat{\mathbf{w}}^t} \frac{1}{2} \sum_{l=1}^{L} \left( \sqrt{\mathbf{d}_l^{t-1}}^{\top} (\hat{\mathbf{w}}_l^t - \mathbf{w}_l^t) \right)^2, \tag{6}$$

*where $\mathbf{d}_l^{t-1} \equiv diag(\mathbf{D}_l^{t-1})$, and*

$$\mathbf{w}_l^t \equiv \hat{\mathbf{w}}_l^{t-1} - \nabla_l\ell(\hat{\mathbf{w}}^{t-1}) \oslash \mathbf{d}_l^{t-1}. \tag{7}$$

Obviously, this objective can be minimized layer by layer. Each proximal Newton iteration thus consists of two steps: (i) Obtain $\mathbf{w}_l^t$ in (7) by gradient descent along $\nabla_l\ell(\hat{\mathbf{w}}^{t-1})$, which is preconditioned by the adaptive learning rate $1 \oslash \mathbf{d}_l^{t-1}$ so that the rescaled dimensions have similar curvatures; (ii) Quantize $\mathbf{w}_l^t$ to $\hat{\mathbf{w}}_l^t$ by minimizing the scaled difference between $\hat{\mathbf{w}}_l^t$ and $\mathbf{w}_l^t$ in (6). Intuitively, when the curvature is low ($[\mathbf{d}_l^{t-1}]_i$ is small), the loss is not sensitive to the weight and ternarization error can be less penalized. When the loss surface is steep, ternarization has to be more accurate.

Though the constraint in (6) is more complicated than that in LAB, interestingly the following simple relationship can still be obtained for weight ternarization.

**Proposition 3.2** *With $\mathcal{Q} = \{-1, 0, 1\}$, and the optimal $\hat{\mathbf{w}}_l^t$ in (6) of the form $\alpha\mathbf{b}$. For a fixed $\mathbf{b}$, $\alpha = \frac{\|\mathbf{b}\odot\mathbf{d}_l^{t-1}\odot\mathbf{w}_l^t\|_1}{\|\mathbf{b}\odot\mathbf{d}_l^{t-1}\|_1}$; whereas when $\alpha$ is fixed, $\mathbf{b} = \mathbf{I}_{\alpha/2}(\mathbf{w}_l^t)$.*

Equivalently, $\mathbf{b}$ can be written as $\Pi_{\mathcal{Q}}(\mathbf{w}_l^t/\alpha)$, where $\Pi_{\mathcal{Q}}(\cdot)$ projects each entry of the input argument to the nearest element in $\mathcal{Q}$. Further discussions on how to solve for $\alpha_l^t$ will be presented in Sections 3.1.1 and 3.1.2. When the curvature is the same for all dimensions at layer $l$, the following Corollary shows that the solution above reduces that of TWN.

**Corollary 3.1** *When $\mathbf{D}_l^{t-1} = \lambda\mathbf{I}$, $\alpha_l^t$ reduces to the TWN solution in (1) with $\Delta_l^t = \alpha_l^t/2$.*

In other words, TWN corresponds to using the proximal gradient algorithm, while the proposed method corresponds to using the proximal Newton algorithm with diagonal Hessian. In composite optimization, it is known that the proximal Newton algorithm is more efficient than the proximal gradient algorithm (Lee et al., 2014; Rakotomamonjy et al., 2016). Moreover, note that the interesting relationship $\Delta_l^t = \alpha_l^t/2$ is not observed in TWN, while TTQ completely neglects this relationship.

In LBNN (Leng et al., 2017), its projection step uses an objective which is similar to (6), but without using the curvature information. Besides, their $\mathbf{w}_l^t$ is updated with the extra-gradient in (2), which doubles the number of forward, backward and update steps, and can be costly. Moreover, LBNN uses full-precision weights in the forward pass, while all other quantization methods including ours use quantized weights (which eliminates most of the multiplications and thus faster training).

When (i) $\ell$ is continuously differentiable with Lipschitz-continuous gradient (i.e., there exists $\beta > 0$ such that $\|\nabla\ell(\mathbf{u}) - \nabla\ell(\mathbf{v})\|_2 \leq \beta\|\mathbf{u} - \mathbf{v}\|_2$ for any $\mathbf{u}, \mathbf{v}$); (ii) $\ell$ is bounded from below; and (iii) $[\mathbf{d}_l^t]_k > \beta \ \forall l, k, t$, it can be shown that the objective of (3) produced by the proximal Newton algorithm (with solution in Proposition 3.2) converges (Hou et al., 2017). In practice, it is important to keep the full-precision weights during update (Courbariaux et al., 2015). Hence, we replace (7) by $\mathbf{w}_l^t \leftarrow \mathbf{w}_l^{t-1} - \nabla_l\ell(\hat{\mathbf{w}}^{t-1}) \oslash \mathbf{d}_l^{t-1}$. The whole procedure, which is called Loss-Aware Ternarization (LAT), is shown in Algorithm 3 of Appendix B. It is similar to Algorithm 1 of LAB (Hou et al., 2017), except that $\alpha_l^t$ and $\mathbf{b}_l^t$ are computed differently. In step 4, following (Li & Liu, 2016), we first rescale input $\mathbf{x}_l^{t-1}$ with $\alpha_l$, so that multiplications in dot products and convolutions become additions. Algorithm 3 can also be easily extended to ternarize weights in recurrent networks. Interested readers are referred to (Hou et al., 2017) for details.

### 3.1.1 Exact solution of $\alpha_l^t$

To simplify notations, we drop the superscripts and subscripts. From Proposition 3.2,

$$\alpha = \frac{\|\mathbf{b} \odot \mathbf{d} \odot \mathbf{w}\|_1}{\|\mathbf{b} \odot \mathbf{d}\|_1}, \quad \mathbf{b} = \mathbf{I}_{\alpha/2}(\mathbf{w}). \tag{8}$$

We now consider how to solve for $\alpha$. First, we introduce some notations. Given a vector $\mathbf{x} = [x_1, x_2, \ldots, x_n]$, and an indexing vector $\mathbf{s} \in \mathbb{R}^n$ whose entries are a permutation of $\{1, \ldots, n\}$, $\text{perm}_{\mathbf{s}}(\mathbf{x})$ returns the vector $[x_{s_1}, x_{s_2}, \ldots x_{s_n}]$, and $\text{cum}(\mathbf{x}) = [x_1, \sum_{i=1}^{2} x_i, \ldots, \sum_{i=1}^{n} x_i]$ returns partial sums for elements in $\mathbf{x}$. For example, let $\mathbf{a} = [1, -1, -2]$, and $\mathbf{b} = [3, 1, 2]$. Then, $\text{perm}_{\mathbf{b}}(\mathbf{a}) = [-2, 1, -1]$ and $\text{cum}(\mathbf{a}) = [1, 0, -2]$.

We sort elements of $|\mathbf{w}|$ in descending order, and let the vector containing the sorted indices be $\mathbf{s}$. For example, if $\mathbf{w} = [1, 0, -2]$, then $\mathbf{s} = [3, 1, 2]$. From (8),

$$\alpha = \frac{\|\mathbf{I}_{\alpha/2}(\mathbf{w}) \odot \mathbf{d} \odot \mathbf{w}\|_1}{\|\mathbf{I}_{\alpha/2}(\mathbf{w}) \odot \mathbf{d}\|_1} = \frac{[\text{cum}(\text{perm}_{\mathbf{s}}(|\mathbf{d} \odot \mathbf{w}|))]_j}{[\text{cum}(\text{perm}_{\mathbf{s}}(|\mathbf{d}|))]_j} = 2c_j, \tag{9}$$

where $\mathbf{c} = \text{cum}(\text{perm}_{\mathbf{s}}(|\mathbf{d} \odot \mathbf{w}|)) \oslash \text{cum}(\text{perm}_{\mathbf{s}}(\mathbf{d})) \oslash 2$, and $j$ is the index such that

$$[\text{perm}_{\mathbf{s}}(|\mathbf{w}|)]_j > c_j > [\text{perm}_{\mathbf{s}}(|\mathbf{w}|)]_{j+1}. \tag{10}$$

For simplicity of notations, let the dimensionality of $\mathbf{w}$ (and thus also of $\mathbf{c}$) be $n$, and the operation $\text{find}(\text{condition}(\mathbf{x}))$ returns all indices in $\mathbf{x}$ that satisfies the condition. It is easy to see that any $j$ satisfying (10) is in $\mathcal{S} \equiv \text{find}([\text{perm}_{\mathbf{s}}(|\mathbf{w}|)]_{[1:(n-1)]} - \mathbf{c}_{[1:(n-1)]}) \odot ([\text{perm}_{\mathbf{s}}(|\mathbf{w}|)]_{[2:n]} - \mathbf{c}_{[1:n-1]}) < 0)$, where $\mathbf{c}_{[1:(n-1)]}$ is the subvector of $\mathbf{c}$ with elements in the index range 1 to $n - 1$. The optimal $\alpha$ $(= 2c_j)$ is then the one which yields the smallest objective in (6), which can be simplified by Proposition 3.3 below. The procedure is shown in Algorithm 1.

**Proposition 3.3** *The optimal $\alpha_l^t$ of (6) equals* $2 \arg\max_{c_j : j \in \mathcal{S}} c_j^2 \cdot [cum(perm_{\mathbf{s}}(\mathbf{d}_l^{t-1}))]_j$.

---

**Algorithm 1** Exact solver of (6)

1: **Input:** full-precision weight $\mathbf{w}_l^t$, diagonal entries of the approximate Hessian $\mathbf{d}_l^{t-1}$.
2: $\mathbf{s} = \arg\text{sort}(|\mathbf{w}_l^t|)$;
3: $\mathbf{c} = \text{cum}(\text{perm}_{\mathbf{s}}(|\mathbf{d}_l^{t-1} \odot \mathbf{w}_l^t|)) \oslash \text{cum}(\text{perm}_{\mathbf{s}}(\mathbf{d}_l^{t-1})) \oslash 2$;
4: $\mathcal{S} = \text{find}(([\text{perm}_{\mathbf{s}}(|\mathbf{w}_l^t|)]_{[1:(n-1)]} - \mathbf{c}_{[1:(n-1)]}) \odot ([\text{perm}_{\mathbf{s}}(|\mathbf{w}_l^t|)]_{[2:n]} - \mathbf{c}_{[1:n-1]}) < 0)$;
5: $\alpha_l^t = 2 \arg\max_{c_j : j \in \mathcal{S}} c_j^2 \cdot [\text{cum}(\text{perm}_{\mathbf{s}}(\mathbf{d}_l^{t-1}))]_j$;
6: $\mathbf{b}_l^t = \mathbf{I}_{\alpha_l^t/2}(\mathbf{w}_l^t)$;
7: **Output:** $\hat{\mathbf{w}}_l^t = \alpha_l^t \mathbf{b}_l^t$.

---

### 3.1.2 Approximate solution of $\alpha_l^t$

In case the sorting operation in step 2 is expensive, $\alpha_l^t$ and $\mathbf{b}_l^t$ can be obtained by alternating the iteration in Proposition 3.2 (Algorithm 2). Empirically, it converges very fast, usually in 5 iterations.

---

**Algorithm 2** Approximate solver for (6).

1: **Input:** $\mathbf{b}_l^{t-1}$, full-precision weight $\mathbf{w}_l^t$, diagonal entries of the approximate Hessian $\mathbf{d}_l^{t-1}$.
2: **Initialize:** $\alpha = 1.0, \alpha_{\text{old}} = 0.0, \mathbf{b} = \mathbf{b}_l^{t-1}, \epsilon = 10^{-6}$;
3: **while** $|\alpha - \alpha_{\text{old}}| > \epsilon$ **do**
4: $\quad \alpha_{\text{old}} = \alpha$;
5: $\quad \alpha = \frac{\|\mathbf{b} \odot \mathbf{d}_l^{t-1} \odot \mathbf{w}_l^t\|_1}{\|\mathbf{b} \odot \mathbf{d}_l^{t-1}\|_1}$;
6: $\quad \mathbf{b} = \mathbf{I}_{\alpha/2}(\mathbf{w}_l^t)$;
7: **end while**
8: **Output:** $\hat{\mathbf{w}}_l^t = \alpha \mathbf{b}$.

---

## 3.2 EXTENSION TO TERNARIZATION WITH TWO SCALING PARAMETERS

As in TTQ (Zhu et al., 2017), we can use different scaling parameters for the positive and negative weights in each layer. The optimization subproblem at the $t$th iteration then becomes:

$$\min_{\hat{\mathbf{w}}^t} \quad \nabla\ell(\hat{\mathbf{w}}^{t-1})^\top(\hat{\mathbf{w}}^t - \hat{\mathbf{w}}^{t-1}) + \frac{1}{2}(\hat{\mathbf{w}}^t - \hat{\mathbf{w}}^{t-1})^\top \mathbf{D}^{t-1}(\hat{\mathbf{w}}^t - \hat{\mathbf{w}}^{t-1}) \tag{11}$$

$$\text{s.t.} \quad \hat{\mathbf{w}}_l^t \in \{-\beta_l^t, 0, \alpha_l^t\}^{n_l}, \ \alpha_l^t > 0, \ \beta_l^t > 0, \ l = 1, \dots, L.$$

**Proposition 3.4** *The optimal $\hat{\mathbf{w}}_l^t$ in (5) is of the form $\hat{\mathbf{w}}_l^t = \alpha_l^t\mathbf{p}_l^t + \beta_l^t\mathbf{q}_l^t$, where $\alpha_l^t = \frac{\|\mathbf{p}_l^t \odot \mathbf{d}_l^{t-1} \odot \mathbf{w}_l^t\|_1}{\|\mathbf{p}_l^t \odot \mathbf{d}_l^{t-1}\|_1}, \mathbf{p}_l^t = \mathbf{I}_{\alpha_l^t/2}^+(\mathbf{w}_l^t), \beta_l^t = \frac{\|\mathbf{q}_l^t \odot \mathbf{d}_l^{t-1} \odot \mathbf{w}_l^t\|_1}{\|\mathbf{q}_l^t \odot \mathbf{d}_l^{t-1}\|_1}, \text{ and } \mathbf{q}_l^t = \mathbf{I}_{\beta_l^t/2}^-(\mathbf{w}_l^t).$*

The exact and approximate solutions for $\alpha_l^t$ and $\beta_l^t$ can be obtained in a similar way as in Sections 3.1.1 and 3.1.2. Details are in Appendix C.

## 3.3 EXTENSION TO LOW-BIT QUANTIZATION

For $m$-bit quantization, we simply change the set $\mathcal{Q}$ of desired quantized values in (3) to one with $k = 2^{m-1} - 1$ quantized values. The optimization still contains a gradient descent step with adaptive learning rates like LAT, and a quantization step which can be solved efficiently by alternating minimization of $(\alpha, \mathbf{b})$ (similar to the procedure in Algorithm 2) using the following Proposition.

**Proposition 3.5** *Let the optimal $\hat{\mathbf{w}}_l^t$ in (6) be of the form $\alpha\mathbf{b}$. For a fixed $\mathbf{b}$, $\alpha = \frac{\|\mathbf{b} \odot \mathbf{d}_l^{t-1} \odot \mathbf{w}_l^t\|_1}{\|\mathbf{b} \odot \mathbf{d}_l^{t-1}\|_1}$; whereas when $\alpha$ is fixed, $\mathbf{b} = \Pi_{\mathcal{Q}}(\frac{\mathbf{w}_l^t}{\alpha})$, where $\mathcal{Q} = \left\{-1, -\frac{k-1}{k}, \dots, -\frac{1}{k}, 0, \frac{1}{k}, \dots, \frac{k-1}{k}, 1\right\}$ for linear quantization and $\mathcal{Q} = \left\{-1, -\frac{1}{2}, \dots, -\frac{1}{2^{k-1}}, 0, \frac{1}{2^{k-1}}, \dots, \frac{1}{2}, 1\right\}$ for logarithmic quantization.*

## 4 EXPERIMENTS

In this section, we perform experiments on both feedforward and recurrent neural networks. The following methods are compared: (i) the original full-precision network; (ii) weight-binarized networks, including BinaryConnect (Courbariaux et al., 2015), Binary-Weight-Network (BWN) (Rastegari et al., 2016), and Loss-Aware Binarized network (LAB) (Hou et al., 2017); (iii) weight-ternarized networks, including Ternary Weight Networks (TWN) (Li & Liu, 2016), Trained Ternary Quantization (TTQ)[2] (Zhu et al., 2017), the proposed Loss-Aware Ternarized network with exact solution (LATe), approximate solution (LATa), and with two scaling parameters (LAT2e and LAT2a); (iv) $m$-bit-quantized networks (where $m > 2$), including DoReFa-Netm (Zhou et al., 2016), the proposed loss-aware quantized network with linear quantization (LAQm(linear)), and logarithmic quantization (LAQm(log)). Since weight quantization can be viewed as a form of regularization (Courbariaux et al., 2015), we do not use other regularizers such as dropout and weight decay.

## 4.1 FEEDFORWARD NETWORKS

In this section, we perform experiments with the multilayer perceptron (on the *MNIST* data set) and convolutional neural networks (on *CIFAR-10*, *CIFAR-100* and *SVHN*). For *MNIST*, *CIFAR-10*, and *SVHN*, the setup is similar to that in (Courbariaux et al., 2015; Hou et al., 2017). Details can be found in Appendix D. For *CIFAR-100*, we use $45,000$ images for training, another $5,000$ for validation, and the remaining $10,000$ for testing. The testing errors are shown in Table 1.

**Ternarization:** On *MNIST*, *CIFAR100* and *SVHN*, the weight-ternarized networks perform better than weight-binarized networks, and are comparable to the full-precision networks. Among the weight-ternarized networks, the proposed LAT and its variants have the lowest errors. On *CIFAR-10*, LATa has similar performance as the full-precision network, but is outperformed by BinaryConnect.

Figure 1(a) shows convergence of the training loss for LATa on *CIFAR-10*, and Figure 1(b) shows the scaling parameter obtained at each CNN layer. As can be seen, the scaling parameters for the

---

[2]For TTQ, we follow the *CIFAR-10* setting in (Zhu et al., 2017), and set $\Delta_l^t = 0.005 \max(|\mathbf{w}_l^t|)$.

Table 1: Testing errors (%) on the feedforward networks. Algorithm with the lowest error in each group is highlighted.

|  |  | MNIST | CIFAR-10 | CIFAR-100 | SVHN |
|---|---|---|---|---|---|
| no binarization | full-precision | 1.11 | 10.38 | 39.06 | 2.28 |
| binarization | BinaryConnect | 1.28 | **9.86** | 46.42 | 2.45 |
|  | BWN | 1.31 | 10.51 | 43.62 | 2.54 |
|  | LAB | **1.18** | 10.50 | **43.06** | **2.35** |
| ternarization | 1 scaling TWN | 1.23 | 10.64 | 43.49 | 2.37 |
|  | 1 scaling LATe | 1.15 | 10.47 | 39.10 | **2.30** |
|  | 1 scaling LATa | **1.14** | **10.38** | 39.19 | **2.30** |
|  | 2 scaling TTQ | 1.20 | 10.59 | 42.09 | 2.38 |
|  | 2 scaling LAT2e | 1.20 | 10.45 | 39.01 | 2.34 |
|  | 2 scaling LAT2a | 1.19 | 10.48 | **38.84** | 2.35 |
| 3-bit quantization | DoReFa-Net3 | 1.31 | 10.54 | 45.05 | 2.39 |
|  | LAQ3(linear) | 1.20 | 10.67 | 38.70 | 2.34 |
|  | LAQ3(log) | **1.16** | **10.52** | **38.50** | **2.29** |

first and last layers (conv1 and linear3, respectively) are larger than the others. This agrees with the finding that, to maintain the activation variance and back-propagated gradients variance during the forward and backward propagations, the variance of the weights between the $l$th and $(l+1)$th layers should roughly follow $2/(n_l + n_{l+1})$ (Glorot & Bengio, 2010). Hence, as the input and output layers are small, larger scaling parameters are needed for their high-variance weights.

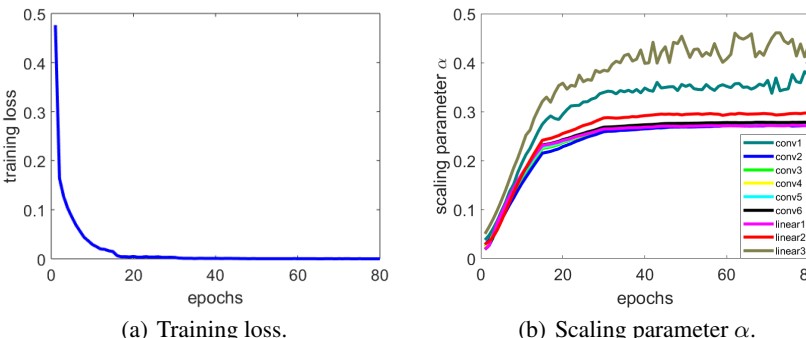

(a) Training loss.          (b) Scaling parameter $\alpha$.

Figure 1: Convergence of the training loss and scaling parameter by LATa on *CIFAR-10*.

**Using Two Scaling Parameters:** Compared to TTQ, the proposed LAT2 always has better performance. However, the extra flexibility of using two scaling parameters does not always translate to lower testing error. As can be seen, it outperforms algorithms with one scaling parameter only on *CIFAR-100*. We speculate this is because the capacities of deep networks are often larger than needed, and so the limited expressiveness of quantized weights may not significantly deteriorate performance. Indeed, as pointed out in (Courbariaux et al., 2015), weight quantization is a form of regularization, and can contribute positively to the performance.

**Using More Bits:** Among the 3-bit quantization algorithms, the proposed scheme with logarithmic quantization has the best performance. It also outperforms the other quantization algorithms on *CIFAR-100* and *SVHN*. However, as discussed above, more quantization flexibility is useful only when the weight-quantized network does not have enough capacity.

## 4.2 Recurrent Networks

In this section, we follow (Hou et al., 2017) and perform character-level language modeling experiments on the long short-term memory (LSTM) (Hochreiter & Schmidhuber, 1997). The training objective is the cross-entropy loss over all target sequences. Experiments are performed on three

data sets: (i) Leo Tolstoy's *War and Peace*; (ii) source code of the *Linux Kernel*; and (iii) *Penn Treebank* Corpus (Taylor et al., 2003). For the first two, we follow the setting in (Karpathy et al., 2016; Hou et al., 2017). For *Penn Treebank*, we follow the setting in (Mikolov & Zweig, 2012). In the experiment, we tried different initializations for TTQ and then report the best. Cross-entropy values on the test set are shown in Table 2.

Table 2: Testing cross-entropy values on the LSTM. Algorithm with the lowest cross-entropy value in each group is highlighted.

| | | *War and Peace* | *Linux Kernel* | *Penn Treebank* |
|---|---|---|---|---|
| no binarization | full-precision | 1.268 | 1.326 | 1.083 |
| binarization | BinaryConnect | 2.942 | 3.532 | 1.737 |
| | BWN | 1.313 | 1.307 | **1.078** |
| | LAB | **1.291** | **1.305** | 1.081 |
| ternarization | TWN | 1.290 | 1.280 | 1.045 |
| 1 scaling | LATe | 1.248 | **1.256** | 1.022 |
| | LATa | 1.253 | 1.264 | 1.024 |
| 2 scaling | TTQ | 1.272 | 1.302 | 1.031 |
| | LAT2e | 1.239 | 1.258 | 1.018 |
| | LAT2a | **1.245** | 1.258 | **1.015** |
| 3-bit quantization | DoReFa-Net3 | 1.349 | 1.276 | 1.017 |
| | LAQ3(linear) | 1.282 | 1.327 | 1.017 |
| | LAQ3(log) | **1.268** | **1.273** | **1.009** |
| 4-bit quantization | DoReFa-Net4 | 1.328 | 1.320 | 1.019 |
| | LAQ4 (linear) | 1.294 | 1.337 | 1.046 |
| | LAQ4 (log) | **1.272** | **1.319** | **1.016** |

**Ternarization:** As in Section 4.1, the proposed LATe and LATa outperform the other weight ternarization schemes, and are even better than the full-precision network on all three data sets. Figure 2 shows convergence of the training and validation losses on *War and Peace*. Among the ternarization methods, LAT and its variants converge faster than both TWN and TTQ.

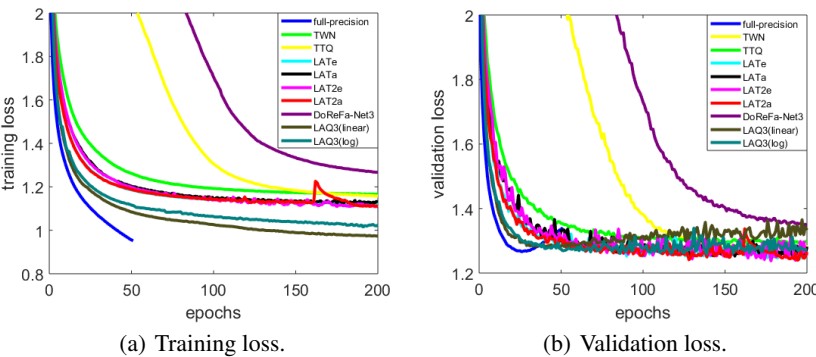

(a) Training loss.        (b) Validation loss.

Figure 2: Convergence of the training and validation losses on *War and Peace*.

**Using Two Scaling Parameters:** LAT2e and LAT2a outperform TTQ on all three data sets. They also perform better than using one scaling parameter on *War and Peace* and *Penn Treebank*.

**Using More Bits:** The proposed LAQ always outperforms DoReFa-Net when 3 or 4 bits are used. As noted in Section 4.1, using more bits does not necessarily yield better generalization performance, and ternarization (using 2 bits) yields the lowest validation loss on *War and Peace* and *Linux Kernel*. Moreover, logarithmic quantization is better than linear quantization. Figure 3 shows distributions of the input-to-hidden (full-precision and quantized) weights of the input gate trained after 20 epochs using LAQ3(linear) and LAQ3(log) (results on the other weights are similar). As can be seen, distributions of the full-precision weights are bell-shaped. Hence, logarithmic quantization can give finer resolutions to many of the weights which have small magnitudes.

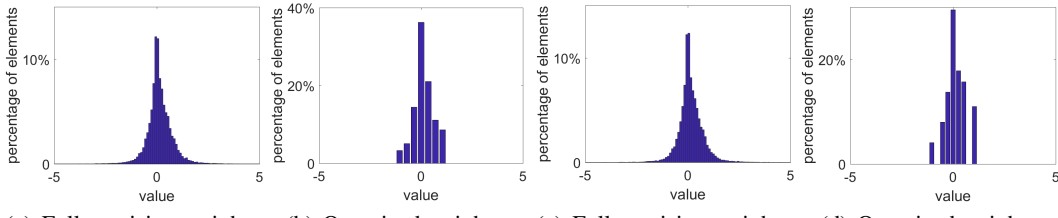

(a) Full-precision weights.   (b) Quantized weights.   (c) Full-precision weights.   (d) Quantized weights.

Figure 3: Distributions of the full-precision and LAQ3-quantized weights on *War and Peace*. Left ((a) and (b)): Linear quantization; Right ((c) and (d)): Logarithmic quantization.

**Quantized vs Full-precision Networks:** The quantized networks often perform better than the full-precision networks. We speculate that this is because deep networks often have larger-than-needed capacities, and so are less affected by the limited expressiveness of quantized weights. Moreover, low-bit quantization acts as regularization, and so contributes positively to the performance.

## 5   CONCLUSION

In this paper, we proposed a loss-aware weight quantization algorithm that directly considers the effect of quantization on the loss. The problem is solved using the proximal Newton algorithm. Each iteration consists of a preconditioned gradient descent step and a quantization step that projects full-precision weights onto a set of quantized values. For ternarization, an exact solution and an efficient approximate solution are provided. The procedure is also extended to the use of different scaling parameters for the positive and negative weights, and to $m$-bit (where $m > 2$) quantization. Experiments on both feedforward and recurrent networks show that the proposed quantization scheme outperforms the current state-of-the-art.

ACKNOWLEDGMENTS

This research was supported in part by the Research Grants Council of the Hong Kong Special Administrative Region (Grant 614513). We thank the developers of Theano (Theano Development Team, 2016), Pylearn2 (Goodfellow et al., 2013) and Lasagne. We also thank NVIDIA for the gift of GPU card.

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

## A    PROOFS

### A.1    PROOF OF PROPOSITION 3.1

With $\mathbf{w}_l^t$ in (7), the objective in (5) can be rewritten as

$$\nabla \ell(\hat{\mathbf{w}}^{t-1})^\top (\hat{\mathbf{w}}^t - \hat{\mathbf{w}}^{t-1}) + \frac{1}{2}(\hat{\mathbf{w}}^t - \hat{\mathbf{w}}^{t-1})^\top \mathbf{D}^{t-1}(\hat{\mathbf{w}}^t - \hat{\mathbf{w}}^{t-1})$$

$$= \frac{1}{2}\sum_{l=1}^{L}\left(\sqrt{\mathbf{d}_l^{t-1}}^\top (\hat{\mathbf{w}}_l^t - (\hat{\mathbf{w}}_l^{t-1} - \nabla_l \ell(\hat{\mathbf{w}}^{t-1}) \oslash \mathbf{d}_l^{t-1}))\right)^2 + c_1$$

$$= \frac{1}{2}\sum_{l=1}^{L}\left(\sqrt{\mathbf{d}_l^{t-1}}^\top (\hat{\mathbf{w}}_l^t - \mathbf{w}_l^t)\right)^2 + c_1$$

$$= \frac{1}{2}\sum_{l=1}^{L}\left(\sqrt{\mathbf{d}_l^{t-1}}^\top (\alpha_l^t \mathbf{b}_l^t - \mathbf{w}_l^t)\right)^2 + c_1$$

$$= \frac{1}{2}\sum_{l=1}^{L}\sum_{i=1}^{n_l}[\mathbf{d}_l^{t-1}]_i(\alpha_l^t[\mathbf{b}_l^t]_i - [\mathbf{w}_l^t]_i)^2 + c_1,$$

where $c_1 = -\frac{1}{2}(\sqrt{\mathbf{d}_l^{t-1}}^\top (\nabla_l \ell(\hat{\mathbf{w}}^{t-1}) \oslash \mathbf{d}_l^{t-1}))^2$ is independent of $\alpha_l^t$ and $\mathbf{b}_l^t$.

### A.2    PROOF OF PROPOSITION 3.2

To simplify notations, we drop the subscript and superscript. Considering one particular layer, problem (6) is of the form:

$$\min_{\alpha,\mathbf{b}} \quad \frac{1}{2}\sum_{i=1}^{n} d_i(\alpha b_i - w_i)^2$$
$$\text{s.t.} \quad \alpha > 0, b_i \in \{-1, 0, 1\}.$$

When $\alpha$ is fixed,

$$b_i = \arg\min_{b_i} \frac{1}{2} d_i(\alpha b_i - w_i)^2 = \frac{1}{2} d_i \alpha^2 (b_i - w_i/\alpha)^2 = \mathbf{I}_{\alpha/2}(w_i).$$

When $\mathbf{b}$ is fixed,

$$\alpha = \arg\min_{\alpha} \frac{1}{2}\sum_{i=1}^{n} d_i(\alpha b_i - w_i)^2$$

$$= \arg\min_{\alpha} \frac{1}{2}\|\mathbf{b} \odot \mathbf{b} \odot \mathbf{d}\|_1 \alpha^2 - \|\mathbf{b} \odot \mathbf{d} \odot \mathbf{w}\|_1 \alpha + c_2,$$

$$= \arg\min_{\alpha} \frac{1}{2}\|\mathbf{b} \odot \mathbf{b} \odot \mathbf{d}\|_1 \left(\alpha - \frac{\|\mathbf{b} \odot \mathbf{d} \odot \mathbf{w}\|_1}{\|\mathbf{b} \odot \mathbf{b} \odot \mathbf{d}\|_1}\right)^2 - \frac{1}{2}\frac{\|\mathbf{b} \odot \mathbf{d} \odot \mathbf{w}\|_1^2}{\|\mathbf{b} \odot \mathbf{b} \odot \mathbf{d}\|_1} + c_2$$

$$= \frac{\|\mathbf{b} \odot \mathbf{d} \odot \mathbf{w}\|_1}{\|\mathbf{b} \odot \mathbf{b} \odot \mathbf{d}\|_1}$$

$$= \frac{\|\mathbf{b} \odot \mathbf{d} \odot \mathbf{w}\|_1}{\|\mathbf{b} \odot \mathbf{d}\|_1}.$$

### A.3    PROOF OF COROLLARY 3.1

When $\mathbf{D}_l^{t-1} = \lambda \mathbf{I}$, i.e., the curvature is the same for all dimensions in the $l$th layer, From Proposition 3.2,

$$\alpha_l^t = \frac{\|\mathbf{b} \odot \mathbf{d}_l^{t-1} \odot \mathbf{w}_l^t\|_1}{\|\mathbf{b} \odot \mathbf{d}_l^{t-1}\|_1} = \frac{\|\mathbf{I}_{\alpha_l^t/2}(\mathbf{w}_l^t) \odot \mathbf{w}_l^t\|_1}{\|\mathbf{I}_{\alpha_l^t/2}(\mathbf{w}_l^t)\|_1} = \frac{1}{\|\mathbf{I}_{\Delta_l^t}(\mathbf{w}_l^t)\|_1}\sum_{i:[\mathbf{w}_l^t]_i > \Delta_l^t} |[\mathbf{w}_l^t]_i|,$$

$$\Delta_l^t = \frac{1}{2}\frac{\|\mathbf{I}_{\alpha_l^t/2}\odot\mathbf{w}_l^t\|_1}{\|\mathbf{I}_{\alpha_l^t/2}\|_1} = \arg\max_{\Delta>0}\frac{1}{\|\mathbf{I}_\Delta(\mathbf{w}_l^t)\|_1}\left(\sum_{i:[\mathbf{w}_l^t]_i>\Delta}|[\mathbf{w}_l^t]_i|\right)^2.$$

This is the same as the TWN solution in (1).

## A.4 PROOF OF PROPOSITION 3.3

For simplicity of notations, we drop the subscript and superscript. For each layer, we have an optimization problem of the form

$$\arg\min_\alpha(\sqrt{\mathbf{d}}^\top(\alpha\mathbf{b}-\mathbf{w}))^2$$

$$= \arg\min_\alpha\|\mathbf{b}\odot\mathbf{b}\odot\mathbf{d}\|_1\left(\alpha-\frac{\|\mathbf{b}\odot\mathbf{d}\odot\mathbf{w}\|_1}{\|\mathbf{b}\odot\mathbf{b}\odot\mathbf{d}\|_1}\right)^2 - \frac{\|\mathbf{b}\odot\mathbf{d}\odot\mathbf{w}\|_1^2}{\|\mathbf{b}\odot\mathbf{b}\odot\mathbf{d}\|_1}$$

$$= \arg\min_\alpha\|\mathbf{I}_{\alpha/2}(\mathbf{w})\odot\mathbf{I}_{\alpha/2}(\mathbf{w})\odot\mathbf{d}\|_1\left(\alpha-\frac{\|\mathbf{I}_{\alpha/2}(\mathbf{w})\odot\mathbf{d}\odot\mathbf{w}\|_1}{\|\mathbf{I}_{\alpha/2}(\mathbf{w})\odot\mathbf{I}_{\alpha/2}(\mathbf{w})\odot\mathbf{d}\|_1}\right)^2 - \frac{\|\mathbf{I}_{\alpha/2}(\mathbf{w})\odot\mathbf{I}_{\alpha/2}(\mathbf{w})\odot\mathbf{w}\|_1^2}{\|\mathbf{I}_{\alpha/2}(\mathbf{w})\odot\mathbf{I}_{\alpha/2}(\mathbf{w})\odot\mathbf{d}\|_1}$$

$$= \arg\min_\alpha-\frac{\|\mathbf{I}_{\alpha/2}(\mathbf{w})\odot\mathbf{d}\odot\mathbf{w}\|_1^2}{\|\mathbf{I}_{\alpha/2}(\mathbf{w})\odot\mathbf{d}\|_1},$$

where the second equality holds as $\mathbf{b} = \mathbf{I}_{\alpha/2}(\mathbf{w})$. From (9), we have

$$-\frac{\|\mathbf{I}_{\alpha/2}(\mathbf{w})\odot\mathbf{d}\odot\mathbf{w}\|_1^2}{\|\mathbf{I}_{\alpha/2}(\mathbf{w})\odot\mathbf{d}\|_1}$$

$$= -\frac{\|\mathbf{I}_{\alpha/2}(\mathbf{w})\odot\mathbf{d}\odot\mathbf{w}\|_1}{\|\mathbf{I}_{\alpha/2}(\mathbf{w})\odot\mathbf{d}\|_1}\cdot\frac{\|\mathbf{I}_{\alpha/2}(\mathbf{w})\odot\mathbf{d}\odot\mathbf{w}\|_1}{\|\mathbf{I}_{\alpha/2}(\mathbf{w})\odot\mathbf{d}\|_1}\cdot\|\mathbf{I}_{\alpha/2}(\mathbf{w})\odot\mathbf{d}\|_1$$

$$= -2c_j\cdot 2c_j\cdot[\text{cum}(\text{perm}_\mathbf{s}(\mathbf{d}))]_j$$

$$= -2c_j^2\cdot[\text{cum}(\text{perm}_\mathbf{s}(\mathbf{d}))]_j.$$

Thus, the $\alpha$ that minimizes $(\sqrt{\mathbf{d}}^\top(\alpha\mathbf{b}-\mathbf{w}))^2$ is $\alpha = 2\arg\max_{c_j,j\in\mathcal{S}}c_j^2\cdot[\text{cum}(\text{perm}_\mathbf{s}(\mathbf{d}))]_j$.

## A.5 PROOF FOR PROPOSITION 3.4

For simplicity of notations, we drop the subscript and superscript, and consider the optimization problem:

$$\min_{\alpha,\mathbf{b}} \quad \frac{1}{2}\sum_{i=1}^n d_i(\hat{w}_i - w_i)^2$$

$$\text{s.t.} \quad \hat{w}_i \in \{-\beta, 0, +\alpha\}.$$

Let $f(\hat{w}_i) = (\hat{w}_i - w_i)^2$. Then, $f(\alpha) = (\alpha - w_i)^2$, $f(0) = w_i^2$, and $f(-\beta) = (\beta + w_i)^2$. It is easy to see that (i) if $w_i > \alpha/2$, $f(\alpha)$ is the smallest; (ii) if $w_i < -\beta/2$, $f(-1)$ is the smallest; (iii) if $-\beta/2 \le w_i \le \alpha/2$, $f(0)$ is the smallest. In other words, the optimal $\hat{w}_i$ satisfies

$$\hat{w}_i = \alpha\mathbf{I}_{\alpha/2}^+(w_i) + \beta\mathbf{I}_{\beta/2}^-(w_i),$$

or equivalently, $\hat{\mathbf{w}} = \alpha\mathbf{p} + \beta\mathbf{q}$, where $\mathbf{p} = \mathbf{I}_{\alpha/2}^+(\mathbf{w})$, and $\mathbf{q} = \mathbf{I}_\beta^-(\mathbf{w})$.

Define $\mathbf{w}^+$ and $\mathbf{w}^-$ such that $[\mathbf{w}^+]_i = \begin{cases} w_i & w_i > 0 \\ 0 & \text{otherwise,} \end{cases}$ and $[\mathbf{w}^-]_i = \begin{cases} w_i & w_i < 0 \\ 0 & \text{otherwise.} \end{cases}$ Then,

$$\frac{1}{2}\sum_{i=1}^n d_i(\hat{w}_i - w_i)^2 = \frac{1}{2}\sum_{i=1}^n d_i(\alpha p_i - w_i^+)^2 + \frac{1}{2}\sum_{i=1}^n d_i(\beta q_i - w_i^-)^2. \tag{12}$$

The objective in (12) has two parts, and each part can be viewed as a special case of the ternarization step in Proposition 3.1 (considering only with positive or negative weights). Similar to the proof for Proposition 3.2, we can obtain that the optimal $\hat{\mathbf{w}} = \alpha\mathbf{p} + \beta\mathbf{q}$ satisfies

$$\alpha = \frac{\|\mathbf{p}\odot\mathbf{d}\odot\mathbf{w}\|_1}{\|\mathbf{p}\odot\mathbf{d}\|_1}, \quad \mathbf{p} = \mathbf{I}_{\alpha/2}^+(\mathbf{w}),$$

$$\beta = \frac{\|\mathbf{q}\odot\mathbf{d}\odot\mathbf{w}\|_1}{\|\mathbf{q}\odot\mathbf{d}\|_1}, \quad \mathbf{q} = \mathbf{I}_{\beta/2}^-(\mathbf{w}).$$

## A.6 PROOF OF PROPOSITION 3.5

For simplicity of notations, we drop the subscript and superscript. Since $\frac{1}{2}(\sqrt{\mathbf{d}}^\top(\alpha\mathbf{b} - \mathbf{w}))^2 = \frac{1}{2}\sum_{i=1}^n d_i(\alpha b_i - w_i)^2$ for each layer, we simply consider the optimization problem:

$$\min_{\alpha,\mathbf{b}} \quad \frac{1}{2}\sum_{i=1}^n d_i(\alpha b_i - w_i)^2$$
$$\text{s.t.} \quad \alpha > 0, b_i \in \mathcal{Q}.$$

When $\alpha$ is fixed,

$$b_i = \arg\min_{b_i} \frac{1}{2} d_i(\alpha b_i - w_i)^2 = \frac{1}{2} d_i \alpha^2 (b_i - w_i/\alpha)^2 = \Pi_{\mathcal{Q}}\left(\frac{w_i}{\alpha}\right).$$

When $\mathbf{b}$ is fixed,

$$\begin{aligned}
\alpha &= \arg\min_\alpha \frac{1}{2}\sum_{i=1}^n d_i(\alpha b_i - w_i)^2 \\
&= \arg\min_\alpha \frac{1}{2}\|\mathbf{b}\odot\mathbf{b}\odot\mathbf{d}\|_1 \alpha^2 - \|\mathbf{b}\odot\mathbf{d}\odot\mathbf{w}\|_1 \alpha + c_2 \\
&= \arg\min_\alpha \frac{1}{2}\|\mathbf{b}\odot\mathbf{b}\odot\mathbf{d}\|_1 \left(\alpha - \frac{\|\mathbf{b}\odot\mathbf{d}\odot\mathbf{w}\|_1}{\|\mathbf{b}\odot\mathbf{b}\odot\mathbf{d}\|_1}\right)^2 - \frac{1}{2}\frac{\|\mathbf{b}\odot\mathbf{d}\odot\mathbf{w}\|_1^2}{\|\mathbf{b}\odot\mathbf{b}\odot\mathbf{d}\|_1} \\
&= \frac{\|\mathbf{b}\odot\mathbf{d}\odot\mathbf{w}\|_1}{\|\mathbf{b}\odot\mathbf{b}\odot\mathbf{d}\|_1} \\
&= \frac{\|\mathbf{b}\odot\mathbf{d}\odot\mathbf{w}\|_1}{\|\mathbf{b}\odot\mathbf{d}\|_1}.
\end{aligned}$$

# B LOSS-AWARE TERNARIZATION ALGORITHM (LAT)

The whole procedure of LAT is shown in Algorithm 3.

# C EXACT AND APPROXIMATE SOLUTIONS FOR TERNARIZATION WITH TWO SCALING PARAMETERS

Let there be $n_1$ positive elements and $n_2$ negative elements in $\mathbf{w}_l$. For a $n$-dimensional vector $\mathbf{x} = [x_1, x_2, \ldots, x_n]$, define inverse$(\mathbf{x}) = [x_n, x_{n-1}, \ldots, x_1]$. As is shown in (12), the objective can be separated into two parts, and each part can be viewed as a special case of ternarization step in Proposition 3.1, dealing only with positive or negative weights. Thus the exact and approximate solutions for $\alpha_l^t$ and $\beta_l^t$ can separately be derived in a similar way as that of using one scaling parameter. The exact and approximate solutions for $\alpha_l^t$ and $\beta_l^t$ for layer-$l$ at the $t$th time step are shown in Algorithms 4 and 5.

# D EXPERIMENTAL DETAILS

## D.1 SETUP FOR FEEDFORWARD NETWORKS

The setup for the four data sets are as follows:

1. *MNIST*: This contains $28 \times 28$ gray images from 10 digit classes. We use $50,000$ images for training, another $10,000$ for validation, and the remaining $10,000$ for testing. We use the 4-layer model:

$$784FC - 2048FC - 2048FC - 2048FC - 10SVM,$$

where $FC$ is a fully-connected layer, and $SVM$ is a $\ell_2$-SVM output layer using the square hinge loss. Batch normalization with a minibatch size 100, is used to accelerate learning. The maximum number of epochs is 50. The learning rate starts at 0.01, and decays by a factor of 0.1 at epochs 15 and 25.

---

**Algorithm 3** Loss-Aware Ternarization (LAT) for training a feedforward neural network.

---

**Input:** Minibatch $\{(\mathbf{x}_0^t, \mathbf{y}^t)\}$, current full-precision weights $\{\mathbf{w}_l^t\}$, first moment $\{\mathbf{m}_l^{t-1}\}$, second moment $\{\mathbf{v}_l^{t-1}\}$, and learning rate $\eta^t$.

1: **Forward Propagation**
2: **for** $l = 1$ to $L$ **do**
3:     compute $\alpha_l^t$ and $\mathbf{b}_l^t$ using Algorithm 1 or 2;
4:     rescale the layer-$l$ input: $\tilde{\mathbf{x}}_{l-1}^t = \alpha_l^t \mathbf{x}_{l-1}^t$;
5:     compute $\mathbf{z}_l^t$ with input $\tilde{\mathbf{x}}_{l-1}^t$ and binary weight $\mathbf{b}_l^t$;
6:     apply batch-normalization and nonlinear activation to $\mathbf{z}_l^t$ to obtain $\mathbf{x}_l^t$;
7: **end for**
8: compute the loss $\ell$ using $\mathbf{x}_L^t$ and $\mathbf{y}^t$;
9: **Backward Propagation**
10: initialize output layer's activation's gradient $\frac{\partial \ell}{\partial \mathbf{x}_L^t}$;
11: **for** $l = L$ to 2 **do**
12:     compute $\frac{\partial \ell}{\partial \mathbf{x}_{l-1}^t}$ using $\frac{\partial \ell}{\partial \mathbf{x}_l^t}$, $\alpha_l^t$ and $\mathbf{b}_l^t$;
13: **end for**
14: **Update parameters using Adam**
15: **for** $l = 1$ to $L$ **do**
16:     compute gradients $\nabla_l \ell(\hat{\mathbf{w}}^t)$ using $\frac{\partial \ell}{\partial \mathbf{x}_l^t}$ and $\mathbf{x}_{l-1}^t$;
17:     update first moment $\mathbf{m}_l^t = \beta_1 \mathbf{m}_l^{t-1} + (1 - \beta_1)\nabla_l \ell(\hat{\mathbf{w}}^t)$;
18:     update second moment $\mathbf{v}_l^t = \beta_2 \mathbf{v}_l^{t-1} + (1 - \beta_2)(\nabla_l \ell(\hat{\mathbf{w}}^t) \odot \nabla_l \ell(\hat{\mathbf{w}}^t))$;
19:     compute unbiased first moment $\hat{\mathbf{m}}_l^t = \mathbf{m}_l^t / (1 - \beta_1^t)$;
20:     compute unbiased second moment $\hat{\mathbf{v}}_l^t = \mathbf{v}_l^t / (1 - \beta_2^t)$;
21:     compute current curvature matrix $\mathbf{d}_l^t = \frac{1}{\eta^t}\left(\epsilon \mathbf{1} + \sqrt{\hat{\mathbf{v}}_l^t}\right)$;
22:     update full-precision weights $\mathbf{w}_l^{t+1} = \mathbf{w}_l^t - \hat{\mathbf{m}}_l^t \oslash \mathbf{d}_l^t$;
23:     update learning rate $\eta^{t+1} = \text{UpdateLearningrate}(\eta^t, t + 1)$;
24: **end for**

---

**Algorithm 4** Exact solver for $\tilde{\mathbf{w}}_l^t$ with two scaling parameters.

---

1: **Input:** full-precision weight $\mathbf{w}_l^t$, diagonal entries of the approximate Hessian $\mathbf{d}_l^{t-1}$.
2: $\mathbf{s}_1 = \arg\text{sort}(\mathbf{w}_l^t)$;
3: $\mathbf{c}_1 = \text{cum}(\text{perm}_{\mathbf{s}_1}(|\mathbf{d}_l^{t-1} \odot \mathbf{w}_l^t|)) \oslash \text{cum}(\text{perm}_{\mathbf{s}_1}(|\mathbf{d}_l^{t-1}|)) \oslash 2$;
4: $\mathcal{S}_1 = \text{find}[([\text{perm}_{\mathbf{s}_1}(\mathbf{w}_l^t)]_{[1:(n_1-1)]} - [\mathbf{c}_1]_{[1:(n_1-1)]}) \odot [\text{perm}_{\mathbf{s}_1}(\mathbf{w}_l^t)]_{[2:n_1]} - [\mathbf{c}_1]_{[1:n_1-1]}) < 0)$;
5: $\alpha_l^t = 2\arg\max_{c_i, i \in \mathcal{S}_1}[\mathbf{c}_1]_i^2 \cdot [\text{cum}(\text{perm}_{\mathbf{s}_1}(|\mathbf{d}_l^{t-1}|))]_i$;
6: $\mathbf{p}_l^t = \mathbf{I}_{\alpha/2}^+(\mathbf{w}_l^t)$;
7: $\mathbf{s}_2 = \text{inverse}(\mathbf{s}_1)$;
8: $\mathbf{c}_2 = \text{cum}(\text{perm}_{\mathbf{s}_2}(|\mathbf{d}_l^{t-1} \odot \mathbf{w}_l^t|)) \oslash \text{cum}(\text{perm}_{\mathbf{s}_2}(|\mathbf{d}_l^{t-1}|)) \oslash 2$;
9: $\mathcal{S}_2 = \text{find}(([-\text{perm}_{\mathbf{s}_2}(\mathbf{w}_l^t)]_{[1:(n_2-1)]} - [\mathbf{c}_2]_{[1:(n_2-1)]}) \odot ([-\text{perm}_{\mathbf{s}_2}(\mathbf{w}_l^t)]_{[2:n_2]} - [\mathbf{c}_2]_{[1:n_2-1]}) < 0)$;
10: $\beta_l^t = 2\arg\max_{c_i, i \in \mathcal{S}_2}[\mathbf{c}_2]_i^2 \odot [\text{cum}(\text{perm}_{\mathbf{s}_2}(|\mathbf{d}_l^{t-1}|))]_i$;
11: $\mathbf{q}_l^t = \mathbf{I}_{\beta/2}^-(\mathbf{w}_l^t)$;
12: **Output:** $\hat{\mathbf{w}}_l^t = \alpha_l^t \mathbf{p}_l^t + \beta_l^t \mathbf{q}_l^t$.

---

2. *CIFAR-10*: This contains $32 \times 32$ color images from 10 object classes. We use $45,000$ images for training, another $5,000$ for validation, and the remaining $10,000$ for testing. The images are preprocessed with global contrast normalization and ZCA whitening. We use the VGG-like architecture:

$$(2 \times 128C3) - MP2 - (2 \times 256C3) - MP2 - (2 \times 512C3) - MP2 - (2 \times 1024FC) - 10SVM,$$

where $C3$ is a $3 \times 3$ ReLU convolution layer, and $MP2$ is a $2 \times 2$ max-pooling layer. Batch normalization with a minibatch size of $50$, is used. The maximum number of epochs

---

**Algorithm 5** Approximate solver for $\hat{\mathbf{w}}_l^t$ with two scaling parameters

---

1: **Input:** $\mathbf{b}_l^{t-1}$, full-precision weight $\mathbf{w}_l^t$, and diagonal entries of approximate Hessian $\mathbf{d}_l^{t-1}$.
2: **Initialize:** $\alpha = 1.0, \alpha_{\text{old}} = 0.0, \beta = 1.0, \beta_o = 0.0, \mathbf{b} = \mathbf{b}_l^{t-1}, \mathbf{p} = \mathbf{I}_0^+(\mathbf{b}), \mathbf{q} = \mathbf{I}_0^-(\mathbf{b}), \epsilon = 10^{-6}$.
3: **while** $|\alpha - \alpha_{\text{old}}| > \epsilon$ **and** $|\beta - \beta_{\text{old}}| > \epsilon$ **do**
4: $\quad \alpha_{\text{old}} = \alpha, \beta_{\text{old}} = \beta$;
5: $\quad \alpha = \frac{\|\mathbf{p} \odot \mathbf{d}_l^{t-1} \odot \mathbf{w}_l^t\|_1}{\|\mathbf{p} \odot \mathbf{d}_l^{t-1}\|_1}$;
6: $\quad \mathbf{p} = \mathbf{I}_{\alpha/2}^+(\mathbf{w}_l^t)$;
7: $\quad \beta = \frac{\|\mathbf{q} \odot \mathbf{d}_l^{t-1} \odot \mathbf{w}_l^t\|_1}{\|\mathbf{q} \odot \mathbf{d}_l^{t-1}\|_1}$;
8: $\quad \mathbf{q} = \mathbf{I}_{\beta/2}^-(\mathbf{w}_l^t)$;
9: **end while**
10: **Output:** $\hat{\mathbf{w}}_l^t = \alpha\mathbf{p} + \beta\mathbf{q}$.

---

       is 200. The learning rate for the weight-binarized network starts at 0.03 while for all the other networks starts at 0.002, and decays by a factor of 0.5 after every 15 epochs.

3. *CIFAR-100*: This contains $32 \times 32$ color images from 100 object classes. We use $45,000$ images for training, another $5,000$ for validation, and the remaining $10,000$ for testing. The images are preprocessed with global contrast normalization and ZCA whitening. We use the VGG-like architecture:

$$(2 \times 128C3) - MP2 - (2 \times 256C3) - MP2 - (2 \times 512C3) - MP2 - (2 \times 1024FC) - 100SVM.$$

Batch normalization with a minibatch size of 100, is used. The maximum number of epochs is 200. The learning rate starts at 0.0005, and decays by a factor of 0.5 after every 15 epochs.

4. *SVHN*: This contains $32 \times 32$ color images from 10 digit classes. We use $598,388$ images for training, another $6,000$ for validation, and the remaining $26,032$ for testing. The images are preprocessed with global and local contrast normalization. The model used is:

$$(2 \times 64C3) - MP2 - (2 \times 128C3) - MP2 - (2 \times 256C3) - MP2 - (2 \times 1024FC) - 10SVM.$$

Batch normalization with a minibatch size of 50, is used. The maximum number of epochs is 50. The learning rate starts at 0.001 for the weight-binarized network, and 0.0005 for the other networks. It then decays by a factor of 0.1 at epochs 15 and 25.

### D.2 SETUP FOR RECURRENT NETWORKS

The setup for the three data sets are as follows:

1. Leo Tolstoy's *War and Peace*: It consists of 3258K characters of almost entirely English text with minimal markup and a vocabulary size of 87. We use the same training/validation/test set split as in (Karpathy et al., 2016; Hou et al., 2017).

2. The source code of the *Linux Kernel*: This consists of 621K characters and a vocabulary size of 101. We use the same training/validation/test set split as in (Karpathy et al., 2016; Hou et al., 2017).

3. The *Penn Treebank* data set (Taylor et al., 2003): This has been frequently used for language modeling. It contains 50 different characters, including English characters, numbers, and punctuations. We follow the setting in (Mikolov & Zweig, 2012), with 5,017K characters for training, 393K for validation, and 442K characters for testing.

We use a one-layer LSTM with 512 cells. The maximum number of epochs is 200, and the number of time steps is 100. The initial learning rate is 0.002. After 10 epochs, it is decayed by a factor of 0.98 after each epoch. The weights are initialized uniformly in $[0.08, 0.08]$. After each iteration, the gradients are clipped to the range $[-5, 5]$. All the updated weights are clipped to $[-1, 1]$ for binarization and ternarization methods, but not for $m$-bit (where $m > 2$) quantization methods.

