# OpenReview forum: "Loss-aware Weight Quantization of Deep Networks"
_ICLR.cc/2018/Conference — Accept (Poster)_

### Official Review · AnonReviewer3 · 2017-11-24
**Ternarization-based network compression via loss-aware minimization**

**Rating:** 8
**Confidence:** 3

**Review:**

In this paper, the authors propose a method of compressing network by means of weight ternarization. The network weights ternatization is formulated in the form of loss-aware quantization, which originally proposed by Hou et al. (2017).

To this reviewer’s understanding, the proposed method can be regarded as the extension of the previous work of LAB and TWN, which can be the main contribution of the work.

While the proposed method achieved promising results compared to the competing methods, it is still necessary to compare their computational complexity, which is one of the main concerns in network compression.

It would be appreciated to have discussion on the results in Table 2, which tells that the performance of quantized networks is better than the full-precision network.

---

> ### Author Response · Authors · 2017-12-20
> **RE: AnonReviewer3's review**
>
> Thanks for your review and suggestions.
>
> 1. "compare their computational complexity"
>
> - For space (assuming that the weight values are stored in 32 bits for full-precision networks), the memory required by ternarized networks are 16 times smaller than the the full-precision network; while m-bit networks are 32/m times smaller.
> - For time, consider the product WX between a rxs weight matrix W and sxn input matrix X. For full-precision networks, the cost of WX is (M+A)rsn, where M and A are the computation costs of 32-bit floating-point multiplication and addition respectively. With the proposed ternarization, WX is computed by steps 3-5 in Algorithm 3. For illustration, we use the approximate solver (Algorithm 2) to compute the scaling parameter \alpha and ternarized value b (invoked in Step 3 of Algorithm 3). With fixed b, computing \alpha takes 2rs multiplications and 2rs additions. With fixed \alpha, computing b takes rs comparisons. Assume that alternating minimization is run for k steps (empirically, k<=10), the computation cost of ternarization using Algorithm 2 is k(2M+2A+U)rs, where U is the computation cost of 32-bit floating-point comparison. Moreover, Steps 4 and 5 of Algorithm 3 take sn multiplications and rsn additions respectively. Thus the total cost for the product is Arsn + Msn + k(2M+2A+U)rs, and the speedup ratio is S = ((M+A) rsn)/(Arsn + Msn + k(2M+2A+U)rs), which is approximately (M+A)/A (some terms can be omitted as usually r >> 1 and n >>1, and k is very small). Following (Hubara et al, 2016), we consider the implication on power in 45nm technology (with A = 0.9J, and M=3.7J). Substituting into the ratio above, the energy reduction is then approximately 5. For the other ternarization algorithms such as TWN and TTQ, they also need at least sn multiplications and rsn additions for the product of WX, and thus the computation cost is similar to the proposed LAT_a.
> - Details and complexity analysis for the other models will be provided in the final version of the paper.
>
> 2. "discussion on the results in Table 2"
>
> - The quantized LSTM performs better than full-precision network because deep networks often have larger-than-needed capacities, and so are less affected by the limited expressiveness of quantized weights. Besides, low-bit quantization acts as regularization, and so contributes positively to the performance. We will add the discussion in the final version of the paper.

---

### Official Review · AnonReviewer2 · 2017-11-26
**A new method for weight quantization. A step in the right direction, with interesting results, but not a huge level of novelty.**

**Rating:** 6
**Confidence:** 4

**Review:**

This paper proposes a new method to train DNNs with quantized weights, by including the quantization as a constraint in a proximal quasi-Newton algorithm, which simultaneously learns a scaling for the quantized values (possibly different for positive and negative weights).

The paper is very clearly written, and the proposal is very well placed in the context of previous methods for the same purpose. The experiments are very clearly presented and solidly designed.

In fact, the paper is a somewhat simple extension of the method proposed by Hou, Yao, and Kwok (2017), which is where the novelty resides. Consequently, there is not a great degree of novelty in terms of the proposed method, and the results are only slightly better than those of previous methods.

Finally, in terms of analysis of the algorithm, the authors simply invoke a theorem from Hou, Yao, and Kwok (2017), which claims convergence of the proposed algorithm. However, what is shown in that paper is that the sequence of loss function values converges, which does not imply that the sequence of weight estimates also converges, because of the presence of a non-convex constraint ($b_j^t \in Q^{n_l}$). This may not be relevant for the practical results, but to be accurate, it can't be simply stated that the algorithm converges, without a more careful analysis.

---

> ### Author Response · Authors · 2017-12-20
> **RE: AnonReviewer2's review**
>
> Thanks for your review and suggestions.
>
> 1. "the paper is a somewhat simple extension of the method proposed by Hou, Yao, and Kwok (2017), which is where the novelty resides. Consequently, there is not a great degree of novelty in terms of the proposed method"
>
> - Please see our reply to reviewer 1 above.
>
> 2. "the results are only slightly better than those of previous methods"
>
> - The testing errors on these data sets are often only a few percent, and so the improvements may appear small. To have a clearer comparison, we added the percentage degradation of classification error as compared to the full-precision network in Table 1 (https://www.dropbox.com/s/miquko7qhff9kns/iclr2018_rebuttal.pdf?dl=0).
> - As can be seen, among the weight-ternarized networks, the proposed LAT and its variants achieve much smaller performance degradation on all four data sets. Existing methods often have large degradation, while ours has <3% degradation on MNIST and <1% on the other three data sets. On CIFAR-100, the proposed LAT and its variants achieve even better results than the full-precision network.
> - For recurrent networks, we similarly added the percentage degradation of cross-entropy in Table 2. As can be seen, the proposed weight ternarization is the only method that performs even better than the full-precision counterpart on all three data sets. On the Linux Kernel and Penn Treebank data sets, the proposed LAT and its variants even have >5% performance gain. On the War and Peace dataset, the proposed LAT and its variants are the only methods that achieve significantly better results than the full-precision network.
>
> 3. "it can't be simply stated that the algorithm converges"
>
> - On the theory side, we can only show convergence of the objective value. We will clarify this in the final version of the paper.
> - Empirically, the quantized weight also converges, as can be seen from the convergence of $\alpha$ in Figure 1(b).

---

### Official Review · AnonReviewer1 · 2017-11-28
**Novelty**

**Rating:** 6
**Confidence:** 4

**Review:**

This paper extends the loss-aware weight binarization scheme to ternarization and arbitrary m-bit quantization and demonstrate its promising performance in the experiments.

Review:

Pros
This paper formulates the weight quantization of deep networks as an optimization problem in the perspective of loss and solves the problem with a proximal newton algorithm.  They extend the scheme to allow the use of different scaling parameters and to m-bit quantization. Experiments demonstrate the proposed scheme outperforms the state-of-the-art methods.

The experiments are complete and the writing is good.

Cons
Although the work seems convincing, it is a little bit straight-forward derived from the original binarization scheme (Hou et al., 2017) to tenarization or m-bit since there are some analogous extension ideas (Lin et al., 2016b, Li & Liu, 2016b). Algorithm 2 and section 3.2 and 3.3 can be seen as additive complementary.

---

> ### Author Response · Authors · 2017-12-20
> **RE: AnonReviewer1's review**
>
> Thanks for your review and suggestions.
>
> 1. "it is a little bit straight-forward derived from the original binarization scheme (Hou et al., 2017) to ternarization or m-bit"
>
> - While the idea of extending from 1-bit (binarization) to more bits is straightforward, the difficulty and novelty are in the mathematical derivations. In Hou et al. (2017), the optimal closed-form solution for loss-aware binarization can be derived easily. However, for ternarization, the optimal \alpha and b (in Proposition 3.2) cannot be easily solved. A straightforward solution would require combinatorial search. Instead, we proposed an exact solver (Algorithm 1) which relies only on sorting. This can be further simplified to an efficient alternating minimization procedure (Algorithm 2). The same situation applies to m-bit quantization.
>
> 2. "there are some analogous extension ideas (Lin et al., 2016b, Li & Liu, 2016b). Algorithm 2 and section 3.2 and 3.3 can be seen as additive complementary"
>
> - While analogous extension ideas have been proposed, their weight solutions obtained are not rigorously derived. Specifically, ternary-connect (Lin et al., 2016b) performs simple stochastic quantization, but does not relate that to any quality measure (e.g., the loss, or distance between the quantized and full-precision weights). In TWN (Li & Liu, 2016b), obtaining the theoretical optimal solution is time-consuming and so they used a heuristic instead. In this paper, we explicitly consider the quantization effect to the loss (as in Hou et al (2017)). However, the resultant optimization problem is much more difficult than theirs as explained above.

---

### Decision · Program_Chairs · 2018-01-29
**ICLR 2018 Conference Acceptance Decision**

**Decision:**

Accept (Poster)

**Comment:**

While novelty is not the main strength of this paper, there is consensus that presentation is clear and the experimental results are convincing. Given the practical importance of designing and benchmarking methods to compactify deep nets, the paper deserves to be presented at ICLR-2018.